# Generating Prompts in Latent Space for Rehearsal-free Continual Learning

Chengyi Yang
Shanghai Institute of AI for
Education, School of Computer
Science and Technology
East China Normal University
Shanghai, China
52265901027@stu.ecnu.edu.cn

Wentao Liu
Shanghai Institute of AI for
Education, School of Computer
Science and Technology
East China Normal University
Shanghai, China
52275901023@stu.ecnu.edu.cn

Shisong Chen
Shanghai Institute of AI for
Education, School of Computer
Science and Technology
East China Normal University
Shanghai, China
sschen@stu.ecnu.edu.cn

Jiayin Qi
Cyberspace Institute of Advanced
Technology
Guangzhou University
Guangzhou, China
qijiayin@139.com

Aimin Zhou[*]
Shanghai Institute of AI for
Education, School of Computer
Science and Technology
East China Normal University
Shanghai, China
amzhou@cs.ecnu.edu.cn

## Abstract

Continual learning emerges as a framework that trains the model on a sequence of tasks without forgetting previously learned knowledge, which has been applied in multiple multimodal scenarios. Recently, prompt-based continual learning has achieved excellent domain adaptability and knowledge transfer through prompt generation. However, existing methods mainly focus on designing the architecture of a generator, neglecting the importance of providing effective guidance for training the generator. To address this issue, we propose Generating Prompts in Latent Space (GPLS), which considers prompts as latent variables to account for the uncertainty of prompt generation and aligns with the fact that prompts are inserted into the hidden layer outputs and exert an implicit influence on classification. GPLS adopts a trainable encoder to encode task and feature information into prompts with reparameterization technique, and provides refined and targeted guidance for the training process through the evidence lower bound (ELBO) related to Mahalanobis distance. Extensive experiments demonstrate that GPLS achieves state-of-the-art performance on various benchmarks. Our code is available at https://github.com/Hifipsysta/GPLS.

## CCS Concepts

• **Computing methodologies → Lifelong machine learning**; **Learning latent representations**.

[*]Aimin Zhou is the corresponding author.

## Keywords

Continual Learning, Prompts Generation, Variational Inference, Probability Prompt Learning

**ACM Reference Format:**
Chengyi Yang, Wentao Liu, Shisong Chen, Jiayin Qi, and Aimin Zhou. 2024. Generating Prompts in Latent Space for Rehearsal-free Continual Learning. In *Proceedings of the 32nd ACM International Conference on Multimedia (MM '24), October 28-November 1, 2024, Melbourne, VIC, Australia.* ACM, New York, NY, USA, 10 pages. https://doi.org/10.1145/3664647.3681003

## 1 Introduction

Continual learning (CL) aims to mitigate the phenomenon of forgetting old knowledge that occurs when a neural network learns multiple sequentially arriving tasks, which is also known as catastrophic forgetting [11, 29]. Over the past year, continual learning has been applied in multiple multimodal tasks, including cross-modal retrieval [56, 58], visual question answering [23, 37, 59] and visual-language model [60, 61].

In the early stage, researches on CL primarily concentrated on different training strategies, including designing regularization terms for objective function [1, 20, 24, 32, 42], learning representative samples from old tasks repetitively [16, 22, 36, 38, 43, 50], increasing parameters when new tasks arrive [15, 25, 48, 57], and projecting gradients onto the orthogonal direction of old features [5, 10, 26, 27, 39, 40, 46]. The emergence of pre-trained Vision Transformer (ViT) [8] has led to the introduction of prompt-based continual learning [50], which has demonstrated superior performance compared to the above methodologies.

In CL scenarios, the utilization of prompt-based methods provides the advantage of maintaining the frozen state of the backbone parameters throughout the training process. As a result, the backbone parameters remain unchanged from their optimal state for the previous task, thereby mitigating the risk of forgetting, which is a primary concern when learning new tasks. Nonetheless, continual prompt learning encounters challenges related to its dependence on

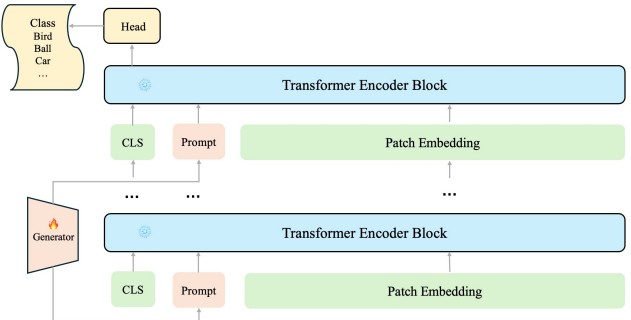

**Figure 1: Illustration of the localization, concentration and indirectness in prompt generation. The number of prompts corresponds to the quantity of transformer blocks, with each prompt being inserted into the input of a distinct transformer layer. While prompt generator is unique and it generates prompts for multiple transformer blocks. Therefore, the trainable parameters are concentrated in a certain area.**

a prompt pool. Specifically, utilizing prompt pool requires setting hyperparameters that are difficult to determine (*e.g.* pool size). More importantly, the number of prompts is commonly much smaller than the number of training instances, which limits the ability of prompts to offer fine-grained guidance to the backbone network.

To avoid dependence on prompt pools, two different solutions have been proposed almost simultaneously: *prompt decomposition* [41] and *prompt generation* [17]. Prompt generation demonstrates better performance in continual learning due to its enhanced domain adaptability. However, we observed that existing prompt generation method lacked direct guidance or constraints on the process of prompt generation. For instance, Domain-Adaptive Prompt (DAP) [17] optimizes the prompt generator just by backpropagating the gradient of final cross-entropy loss, without directly supervising the process of prompt generation. The most direct evidence is that no variables directly related to the prompt generator were considered in the objective function.

However, training a prompt generator only through cross-entropy loss is even more challenging compared to training prompts directly, which is determined by the three major properties of prompt generation methods (also see Figure 1). (1) **Localization**: Cross-entropy serves as a global loss metric, while prompt learning needs to focus on the impact of local parameter variations on classification results, because the backbone parameters are frozen and the trainable generator parameters are concentrated in one or a few narrow regions. (2) **Concentration**: In prompt generation, the concentration of trainable parameters is higher compared to prompt optimization approaches. Specifically, the trainable parameters in prompt optimization methods are distributed across the inputs of various transformer blocks, while the learnable parameters for prompt generation are contained in the unique prompt generator, irrespective of the quantity of transformer blocks utilized. (3) **Indirectness**: Parameter updating in prompt optimization methods can directly affect the classification results, while parameter updating in the prompt generator only exert an indirect influence on classification results via generated prompts.

Regrettably, several studies [2, 13, 30, 54] have shown that parameter sparsity is beneficial for continual learning. In light of these characteristics, we propose Generating Prompts in Latent Space (GPLS) for continual learning. GPLS utilizes an encoder to encode task and feature information into prompts, and provides enhanced and targeted guidance for the training process through variational inference. We conceptualize the generated prompts as observations follow a distribution, which can better describe their uncertainty. Notably, prompts are considered as latent variables as they are inserted into the inputs of different transformer blocks, which is also the hidden layer outputs. Therefore, the problem of prompt generation is reframed as the task of estimating the distribution in latent space. In summary, our main contributions are as follows:

(1) We consider prompts as latent variables following a distribution, which enhances the ability to capture the uncertainty of generated prompts and also conforms to the fact that prompts are inserted into the hidden layer outputs.

(2) We propose a novel method called Generating Prompts in Latent Space (GPLS) for continual learning, which transforms the issue of prompt generation into the distribution estimation in latent space and generates prompts via a trainable encoder with reparameterization technique.

(3) We derive the evidence lower bound (ELBO) for prompt generation and design an objective function related to Mahalanobis distance that offers refined guidance for the encoder to generate better prompts.

(4) Extensive experiments show that GPLS achieves state-of-the-art performance on five standard CL benchmarks, especially on DomainNet with a large number of samples and categories.

## 2 Related Work

### 2.1 Continual Learning

Continual learning aims to mitigate the forgetting of knowledge about old tasks after learning new tasks, which can be categorized into several categories according to the principles of algorithm design. *Regularization-based approaches* [1, 20, 24, 32, 42] mitigate catastrophic forgetting by restricting the parameter updating when the model learns a new task so that it cannot deviate from the optimal parameters on previous tasks. However, these methods also limit the learning capacity of models in the process of constraining parameters. *Expansion-based methods* [15, 25, 48, 57] maintain optimal parameters for old tasks and enhance the plasticity of the model by learning new knowledge through the newly added network parameters. *Replay-based methods* [16, 22, 36, 38, 43, 50] store a portion of previously trained data in a memory buffer and combine it with new training data when learning the new task. Although these two type of approaches achieve better performances, they come with higher training costs and increased graphics memory as the number of tasks grows. *Projection-based methods* [5, 10, 26, 27, 39, 40, 46] project the gradients of current task onto the orthogonal direction of old task features to prevent gradient interference between tasks. However, these methods perform singular value decomposition (SVD) on representation matrices, which becomes ineffective when dealing with large numbers of parameters or training samples.

## 2.2 Prompting for Continual Learning

In contrast to previous approaches, prompt-based CL methods freeze the parameters of backbone, and train prompts inserted into the outputs of hidden layer instead, thus providing adaptation to new tasks. Learning to Prompt (L2P) [50] introduces the concept of prompt pool, which selects prompts with the highest similarity for different inputs. Inspired by complementary learning systems, DualPrompt [49] designs two types of prompts to learn general and specialized knowledge respectively. However, L2P and DualPrompt are inseparable from the step of prompt selection from a prompt pool, which cannot be optimized by backpropagation. For this reason, COntinual Decomposed Attention-based Prompting (CODA-Prompt) [41] introduces prompt decomposition, which decomposes a task-specific prompt into a linear combination of multiple prompts. S-Prompt [47] advocates learning prompts independently according to domains to avoid confusion between new and old knowledge. Language Guidance for prompt-based Continual Learning (LGCL) [18] suggests using natural language prompts to guide the model, while maintaining semantics of prompts also means that the prompt templates are difficult to optimize. Hierarchical Decomposition (HiDe-)Prompt [45] decomposes the objective of prompt learning into task-identity inference, within-task prediction and task-adaptive prediction, which essentially introduces a regularization strategy for prompt-based continual learning.

## 2.3 Prompt Generation

Prompt generation methods train a prompt generator to generate task-specific prompts instead of training the prompts directly. Domain-Adaptive Prompt (DAP) [17] focuses on designing a prompt generator for continual learning, which has better domain adaptability and does not rely on a prompt pool. However, it lacks fine-grained guidance for the training process of its prompt generator. Personalized Federated learning framework of client-specific Prompt Generation (pFedPG) [55] propose a personalized prompt generation method in federated learning scenarios, which is fundamentally different from the continual learning settings we consider. Customized Prompts via Language models (CuPL) [35] generates customized prompt for each class of images in the dataset through GPT-3 [3], while the problem it encounters is the limited control over the generated prompts.

## 2.4 Probability Prompt Learning

Under the perspective of probability prompt learning, deterministic prompts can be viewed as the point estimates of a prompt distribution, which poses challenges in representing the diversity and uncertainty inherently in prompts. PROmpt Distribution leArning (ProDA) [28] designs the first probability model for prompts to estimate the prompt distribution and therefore capture the variability of prompt representations. Prompt Learning with Optimal Transport (PLOT) [6] defines both features and prompt information as discrete distributions under the Dirac measure, and attempts to guide the optimization of prompts through optimal transport theory [44]. Bayesian Prompt Learning (BPL) [7] frame prompt learning from the Bayesian perspective and regularize the prompt space to reduce overfitting on seen prompts. Although these methods also assume prompts as distributions, they are neither continual

learning methods nor prompting generation methods. Therefore, GPLS is fundamentally different from the above methods.

## 3 Method

### 3.1 Background

**Prompt Distribution Learning (ProDA)** [28] is the first method in probability prompt learning, which learns an optimal prompt distribution $p(P)$ targeting at minimizing the empirical loss

$$P^* = \arg\min_{P} \mathbb{E}_{X,Y} \left[ -\log \mathbb{E}_{\theta} p(Y|X,\theta) \right], \tag{1}$$

where $P$ is the prompt. $\theta$ is the normalized embeddings of the text, which is determined by the prompt distribution $p(P)$ and the text encoder $g(\cdot)$. $X$ and $Y$ are the input images and labels respectively. **Bayesian Prompt Learning (BPL)** [7] assumes that the prompt distribution consists of a set of fixed prompts and residuals

$$P(X) = \left[ P_1 + r, P_2 + r, \cdots, P_L + r \right], \ r \sim p(r|X). \tag{2}$$

Here $P_c$ is the fixed prompt for class $c$, and $p(r|X)$ is denoted as the real posterior distribution for $r$. Variational posterior is introduced to approximate the residual distribution, thereby obtaining a lower bound on log-likelihood

$$\log p(Y|X) \geq \mathbb{E}_{p(r|X)} \left[ \log(p(Y|X,r)) \right] - KL(p(r|X)\|p(r)). \tag{3}$$

Aside from the differences in an overall perspective mentioned in Section 2.4, we can also outline three detailed distinctions. (1) We define the prompt distribution by considering prompts as variables that follow a distribution, rather than decompose it into a set of deterministic prompts combined with residual distribution [7]. (2) We train an encoder to generate prompts through a novel variational loss related to Mahalanobis distance, rather than optimize the prompts themselves. (3) We encode task and feature information into task-specific prompts in latent space with the reparameterization technique, rather than training class-specific prompts.

### 3.2 Problem Formulation

In this study, our goal is to generate better prompts $P$ for classification tasks, which is different from previous prompt optimization methods. Instead of training the prompts directly, we focus on training a prompt generator to generate better prompts.

To achieve this goal, we define the prompts as variables that follow a distribution from the perspective of probability prompt learning [28], which offers an effective way to describe the uncertainty of the generated prompts. We denote the prior probability density function of prompt distribution as $p(P)$, where $P$ can be seen as a random variable and its value is an observation. In the training process, both the input data $X$ and labels $Y$ are known. The target is to generate a better prompt $P$ with the help of $X$ and $Y$, which can be described as solving a posterior probability density function $p(P|X,Y)$.

In practice, the value of $P$ can be determined by optimizing a generator with $X$ and $Y$ when freezing the parameters of backbone, but it is still difficult to know the distribution $p(P|X,Y)$ exactly. There are three main reasons for this: (1) Although a specific $P$ can be obtained through optimization, the distribution $p(P|X,Y)$ remains unknown because the value of $P$ is merely an observation. (2) Even though multiple observations of $P$ can be collected

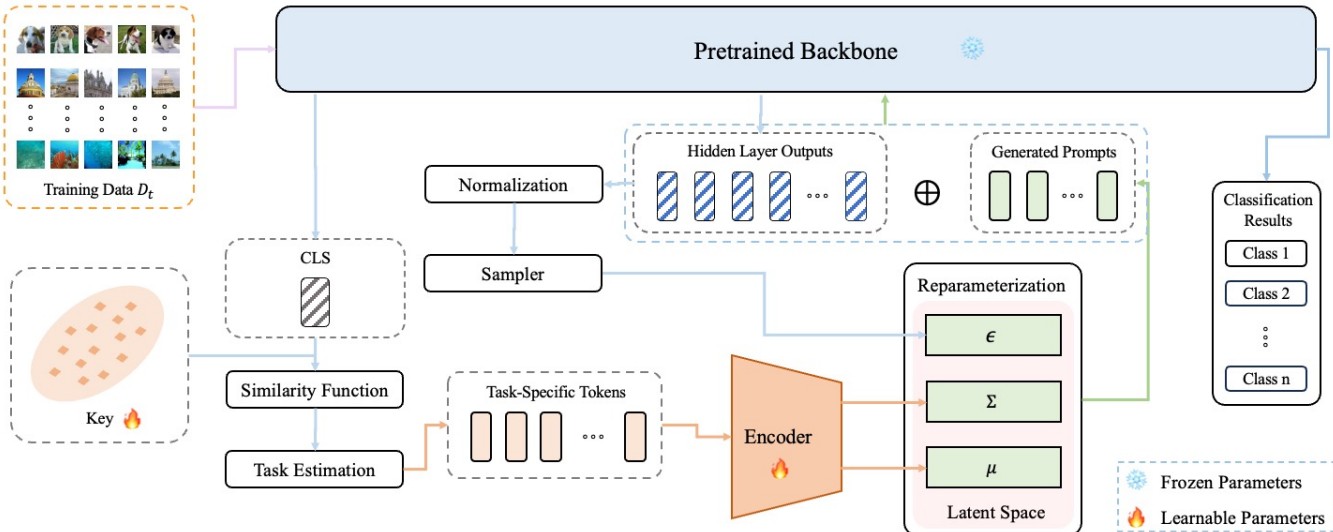

Figure 2: Illustration of the overall framework of GPLS. Firstly, we feed the data into the model and perform forward propagation to obtain the output on position of CLS at the last hidden layer. Secondly, we adopt CLS as the query and train a key to match it. In the training stage, the key is optimized through $\mathcal{L}_{match}$. In the testing stage, task ID is predicted through the maximum similarity between the trainable key and CLS. Then, we establish a set of tokens for each task and input it into an encoder to obtain the mean $\mu$ and covariance $\Sigma$ of the prompt distribution. Another reparameterization variable $\epsilon$ is obtained by normalizing and sampling the inputs of transformer blocks. Finally, the prompts are obtained through reparameterization technique and inserted into the inputs of transformer blocks.

through repeated experiments, they do not represent the ground truth. This is significantly different from the scenarios considered by variational auto-encoder (VAE) [19] and generative adversarial network (GAN) [12], as these methods assume that the observed samples are all real. (3) Although we have defined a symbol for the prior distribution, we do not know the prior information of the prompt distribution. (e.g. The theoretical distribution it follows.) Moreover, since the observed values of $P$ are not the ground truth, it is inappropriate to estimate the prior $p(P)$ through maximum likelihood estimation (MLE) or the posterier $p(P|X,Y)$ through maximum a posteriori (MAP).

### 3.3 Variational Inference in Prompt Generation

From the above analysis, we can summarize that the challenge in estimating the posterior distribution of generated prompts accurately lies in the absence of prior information and observations representing ground truth. To address this issue, we introduce a variational distribution $q(P)$ to approximate the real posterior $p(P|X,Y)$. Since the variational distribution is not real, it can be assumed to follow a certain theoretical distribution.

We hope to find an optimal variational distribution that can be expressed as minimizing the Kullback-Leibler (KL) divergence between $q(P)$ and $p(P|X,Y)$, namely

$$q^*(P) = \underset{q(P)\in\mathbb{Q}}{\arg\min} KL(q(P)\|p(P|X,Y)). \qquad (4)$$

Here $\mathbb{Q}$ is the possible value space for variational distribution $q(P)$. However, we cannot assume to know $Y$ in the testing phase, thus Eq.(4) is not always valid. To address this problem, we regard

prompts as the variable in latent space and expand the logarithmic likelihood probability at the end of the classifier into

$$\log p(Y|X) = \int_P q(P) \log p(Y|X)dP. \qquad (5)$$

After further derivation, we find that the log-likelihood $\log p(Y|X)$ with prompt as the latent variable can be expressed as the sum of evidence lower bound (ELBO) and KL divergence, namely

$$\log p(Y|X) = ELBO(q) + KL(q(P)\|p(P|X,Y)). \qquad (6)$$

Here the specific expressions for ELBO and KL divergence are as follows

$$ELBO(q) = \mathbb{E}_{q(P)}[\log p(P,Y|X)] - \mathbb{E}_{q(P)}[\log q(P)], \qquad (7)$$

$$KL(q(P)\|p(P|X,Y)) = \mathbb{E}_{q(P)}[\log q(P)] - \mathbb{E}_{q(P)}[\log p(P|X,Y)]. \qquad (8)$$

The proof is available at supplementary materials. Since KL divergence is non negative, we can obtain $\log p(Y|X) \geq ELBO(q)$.

Since $\log p(Y|X)$ in Eq.(6) is an observed value, we can transform the objective function in Eq.(4) to maximizing ELBO as

$$\underset{q(P)\in\mathbb{Q}}{\arg\max} \mathbb{E}_{q(P)}[\log p(P,Y|X)] - \mathbb{E}_{q(P)}[\log q(P)], \qquad (9)$$

where $\log p(P,Y|X)$ is the logarithmic likelihood, which can be directly obtained. $\log q(P)$ is the logarithmic variational distribution we are concerned with.

### 3.4 Optimization for Generator

In Section 3.3, we have derived an implicit formulations for the ELBO in prompt generation. Next, we attempt to further provide

its explicit expression. In Eq.(9), $\log p(P, Y|X)$ is computable since it has the following form

$$\log p(P, Y|X) = \log \frac{\exp\left(W_y^\top f(X, P)\right)}{\sum_{c=1}^C \exp\left(W_c^\top f(X, P)\right)}, \quad (10)$$

where $W_y$ and $W_c$ are the weight slices of the last linear layer related to class $y$ and $c$ respectively. $f(X, P)$ is the input of the last linear layer. Prompt $P$ is generated by encoder $\psi(\cdot)$. In addition, $\frac{\exp(\cdot)}{\sum_{c=1}^C \exp(\cdot)}$ is the function of softmax, namely $p(P, Y|X) = \text{softmax}\left(W_y^\top f(X, P)\right)$. Through Monte Carlo estimates of expectations [19], we can obtain the first term of ELBO as

$$\mathbb{E}_{q(P)}[\log p(P, Y|X)] \simeq \frac{1}{M} \sum_{m=1}^M \log\left[\text{softmax}\left(W_y^\top f(X, P_m)\right)\right]. \quad (11)$$

The prompt in each layer is only generated once in each iteration, thus $M = 1$. Since $q(P)$ is not the real distribution of $P$, but an approximate distribution. We can assume that $q(P)$ is a multivariate Gaussian distribution with mean $\mu$ and covariance matrix $\Sigma$ respectively. Its probability density function can be expressed as

$$q(P) = \frac{1}{\sqrt{(2\pi)^k |\Sigma|}} \exp\left(-\frac{1}{2}(P - \mu)^\top \Sigma^{-1}(P - \mu)\right). \quad (12)$$

Here $k$ is the length of the generated prompt, and $|\Sigma|$ represents the determinant of $\Sigma$. We expand the expectation $\mathbb{E}_{q(P)}[\log q(P)]$ into the integral form and further obtain

$$\mathbb{E}_{q(P)}[\log q(P)] = -\frac{k}{2}\log(2\pi) - \frac{1}{2}\log|\Sigma| \\ - \frac{1}{2}\mathbb{E}_{q(P)}\left[(P - \mu)^\top \Sigma^{-1}(P - \mu)\right]. \quad (13)$$

The proof is presented in supplementary materials. Here $(P - \mu)^\top \Sigma^{-1}(P - \mu)$ can be regarded as the square of Mahalanobis distance [51, 53] between $P$ and $\mu$ from a metric learning perspective.

Since prompt length $k$ is a predetermined constant, the first term in Eq.(13) can be ignored during the optimization process. Next, we regard the negative ELBO as the loss function, which can be written in the specific form as

$$\mathcal{L}_{NELBO} = -\mathbb{E}_{q(P)}[\log p(P, Y|X)] + \mathbb{E}_{q(P)}[\log q(P)] \\ = -\log\left[\text{softmax}\left(W_y^\top f(X, P)\right)\right] \\ - \frac{1}{2}\log|\Sigma| - \frac{1}{2}\mathbb{E}_{q(P)}\left[(P - \mu)^\top \Sigma^{-1}(P - \mu)\right]. \quad (14)$$

We also refer to it as variational loss. Intuitively, the target of Eq.(14) is to obtain a better variational distribution for prompts. In practical terms, we optimize an encoder that generates task-specific prompts.

To ensure that the inverse of the covariance matrix can always be obtained, we perform singular value decomposition (SVD) on $\Sigma$ and obtain $\Sigma = U\Lambda V^\top$. Here $U$ and $V$ are left and right singular matrix respectively, and $\Lambda$ is a diagonal matrix with sorted singular values $\lambda_i, \lambda_2, \cdots, \lambda_k$ along its diagonal. Therefore, the inverse matrix of $\Sigma$ can be expressed as

$$\Sigma^{-1} = V\Lambda^{-1}U^\top, \quad (15)$$

because $\Sigma\Sigma^{-1} = U\Lambda V^\top V\Lambda^{-1}U^\top = I$. Note that $V$ and $U$ are both orthogonal matrix according to the properties of SVD. In addition,

since covariance matrix $\Sigma$ is a positive semidefinite matrix, its determinant is non negative. Then $|\Sigma|$ equals the product of singular values for the reason that $|U\Lambda V^\top| = |U| \, |\Lambda| \, |V^\top| = |\Lambda|$, namely $|\Sigma| = |\Lambda| = \lambda_1 \cdot \lambda_2 \cdot \ldots \cdot \lambda_k$.

We denote the second and last term in Eq.(13) as $\mathcal{L}_{nlog\_cov}$ and $\mathcal{L}_{nexp\_Mah}$ respectively, which are the abbreviations of negative logarithmic covariance and negative expected (square of) Mahalanobis distance. Their specific forms can be expressed as

$$\mathcal{L}_{nexp\_Mah} = -\frac{1}{2}\mathbb{E}_{q(P)}\left[(P - \mu)^\top V\Lambda^{-1}U^\top(P - \mu)\right], \quad (16)$$

$$\mathcal{L}_{nlog\_cov} = -\frac{1}{2}\log|\Sigma| = -\frac{1}{2}\sum_{i=1}^k \log \lambda_i. \quad (17)$$

Thus, the final objective function can be expressed as

$$\mathcal{L} = \mathcal{L}_{softmax} + \mathcal{L}_{nlog\_cov} + \mathcal{L}_{nexp\_Mah} + \alpha\mathcal{L}_{match}, \quad (18)$$

where $\mathcal{L}_{softmax} = -\mathbb{E}_{q(P)}[\log p(P, Y|X)]$ is the negative value of Eq(11). $\mathcal{L}_{match}$ is the matching loss applied in L2P [50], Dual-Prompt [49] and DAP [17], which optimizes a query for predicting task ID in class-incremental continual learning.

## 3.5 Geometric Explanation on Variational Loss

Through the above derivation process, we have obtained the explicit expressions of $\mathcal{L}_{nlog\_cov}$ and $\mathcal{L}_{nexp\_Mah}$. Their formulations possess distinct geometric interpretations in the contexts of optimization and metric learning. Specifically, during the backpropagation procedure, the decrease of $\mathcal{L}_{nexp\_Mah}$ corresponds to an elevation in the expectation of squared Mahalanobis distance between the generated prompt and the population moment. When prompt $P$ is allowed to be distant from the population moment in the training process, there is a propensity for $P$ to explore diverse prompts in a wider area in high-dimensional space rather than being confined to a limited region (e.g. around the saddle point). Therefore, the generated prompts are more likely to achieve better domain adaptability. On the other hand, $\mathcal{L}_{nlog\_cov}$ is characterized by the product of eigenvalues, which serves to prevent excessive scaling in the Mahalanobis distance. We explain excess and scaling separately. (1) A small singular value $\lambda_i$ will result in a large value of $-\frac{1}{2}\log \lambda_i$, and further increase the total loss. (2) Mahalanobis distance can obtained from Euclidean distance after rotation and scaling [51, 53]. When covariance matrix $\Sigma$ is an identity matrix, Mahalanobis distance degenerates into Euclidean distance.

## 3.6 Prompt Generation

The prompt in GPLS is generated by reparameterization technique [19], which can be expressed as

$$P = \mu + \Sigma \odot \epsilon, \quad \epsilon \sim p(\epsilon), \quad (19)$$

which ensures that generated prompts $P$ follow the similar distribution with $\epsilon$. We build a sampler to extract the information of hidden layer outputs that generated prompts are inserted into to obtain $\epsilon$.

The mean and covariance matrix of the generated prompts are provided by a trainable encoder $\psi(\cdot)$, namely

$$\mu, \Sigma = \psi(E_t). \quad E_t \in \mathcal{E}. \quad (20)$$

**Table 1: Quantitative results (%) of class-incremental continual learning on Split CIFAR-100 and Split DomainNet. Here BiC, DER++ and DyTox are replay-based methods with a memory buffer of 50 per class, and other methods are all rehearsal-free. † denotes the prompt-based CL methods.**

| Method | Venue | Split CIFAR-100 | | | Split DomainNet | | |
|---|---|---|---|---|---|---|---|
| | | Avg Acc ($\uparrow$) | Lrn Acc ($\uparrow$) | Forgetting ($\downarrow$) | Avg Acc ($\uparrow$) | Lrn Acc ($\uparrow$) | Forgetting ($\downarrow$) |
| EWC [20] | PNAS'17 | 59.60 ± 1.27 | 81.78 ± 1.29 | 24.65 ± 0.07 | 22.35 ± 1.86 | 84.27 ± 2.13 | 64.11 ± 1.28 |
| LwF [24] | TPAMI'17 | 68.22 ± 1.63 | 82.05 ± 0.07 | 15.44 ± 1.48 | 28.86 ± 1.92 | 84.09 ± 1.48 | 56.32 ± 1.01 |
| BiC [52] | CVPR'19 | 81.42 ± 0.85 | 93.37 ± 0.32 | 14.32 ± 1.02 | 68.19 ± 1.22 | 86.61 ± 1.61 | 20.27 ± 0.39 |
| DER++ [4] | NeurIPS'20 | 83.94 ± 0.34 | 91.49 ± 0.61 | 9.87 ± 0.73 | 74.61 ± 0.27 | 88.13 ± 1.14 | 16.05 ± 0.94 |
| DyTox [9] | CVPR'22 | 88.15 ± 0.28 | 90.92 ± 0.78 | 3.64 ± 0.19 | 79.60 ± 0.91 | 84.15 ± 1.18 | 5.87 ± 0.20 |
| L2P† [50] | CVPR'22 | 83.06 ± 0.17 | 88.25 ± 0.01 | 6.58 ± 0.40 | 80.67 ± 0.85 | 85.14 ± 0.99 | 5.33 ± 0.87 |
| DualPrompt† [49] | ECCV'22 | 86.60 ± 0.19 | 90.64 ± 0.01 | 4.45 ± 0.16 | 81.89 ± 0.63 | 87.27 ± 1.80 | 5.21 ± 1.17 |
| S-Prompt† [47] | NeurIPS'22 | 88.81 ± 0.18 | 92.25 ± 0.03 | 3.87 ± 0.05 | 82.15 ± 0.47 | 87.14 ± 0.74 | 5.03 ± 0.22 |
| ESN [48] | AAAI'23 | 86.34 ± 0.52 | 88.92 ± 0.78 | 4.76 ± 0.14 | 68.76 ± 0.12 | 73.23 ± 1.64 | 5.75 ± 0.23 |
| CODA-Prompt† [41] | CVPR'23 | 86.94 ± 0.63 | 91.57 ± 0.75 | 4.04 ± 0.18 | 82.68 ± 0.14 | 87.50 ± 0.08 | 5.29 ± 0.05 |
| HiDe-Prompt† [45] | NeurIPS'23 | 92.61 ± 0.28 | 94.03 ± 0.01 | 3.16 ± 0.10 | 83.16 ± 0.32 | 87.63 ± 0.27 | 4.72 ± 0.46 |
| DAP† [17] | ICCV'23 | 94.05 ± 1.19 | 96.37 ± 0.74 | 2.28 ± 0.96 | 83.51 ± 1.07 | 88.77 ± 0.79 | 5.30 ± 0.52 |
| GPLS† | Ours | **96.22 ± 0.43** | **97.12 ± 0.51** | **1.12 ± 0.32** | **90.13 ± 1.01** | **93.44 ± 0.63** | **3.56 ± 0.49** |

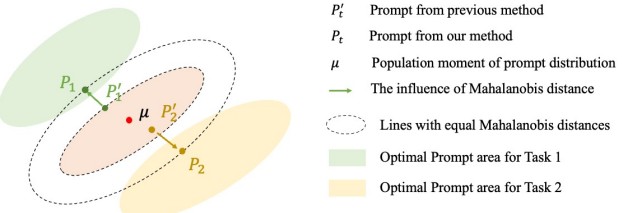

$P'_t$   Prompt from previous method
$P_t$   Prompt from our method
$\mu$   Population moment of prompt distribution
→   The influence of Mahalanobis distance
, ⟩   Lines with equal Mahalanobis distances
  Optimal Prompt area for Task 1
  Optimal Prompt area for Task 2

**Figure 3: Geometric explanation for $\mathcal{L}_{nexp\_Mah}$. In GPLS, prompt encoder is encouraged to generate prompts far from the population moment $\mu$ for different tasks, which enables prompts to obtain better domain adaptivity. The degree of a prompt away from $\mu$ is measured by the square of Mahalanobis distance.**

Here $\mathcal{E}$ is the possible value space for the input of the encoder. $E_t$ is composed of a set of tokens containing task information, which is similar to the text embedding converted from natural language. We establish a key corresponding to each $E_t$, and train it through a query-key matching form

$$t = \arg\max_t \cos(CLS^L, K_t). \quad (21)$$

Here $K_t$ is the learnable key at task $t$, which matchs the CLS of last transformer layer as the query to predict the task ID in testing stage under class-increment settings.

### 3.7 Prompt-based Continual Learning in GPLS

Next, we will illustrate how the generated prompts are applied to a Vision Transformer (ViT) backbone. A typical ViT architecture comprises a patch embedding layer along with several transformer layers. When receiving an input image $X \in \mathbb{R}^{d_H \times d_W \times d_C}$,

the patch embedding layer initially divides the image into a sequence of flattened patches $X_{pat} = \left[ x_{pat,1}, x_{pat,2}, \cdots, x_{pat,d_N} \right] \in \mathbb{R}^{d_N \times (d_P^2 \times d_C)}$. Here $d_H$ and $d_W$ are the height and width of an image, and $d_C$ is the number of input channels. $d_P \times d_P$ is the resolution of each patch. The quantity of patches can be computed as $d_N = \lfloor \frac{d_H}{d_P} \rfloor \times \lfloor \frac{d_W}{d_P} \rfloor$. The input of the first transformer layer can be represented as

$$\left[ CLS^0, P^0, Z^0 \right] = \left[ CLS^0, P^0, X_{pat}W_{emb} + X_{pos} \right], \quad (22)$$

where $CLS$ is a learnable class token vector prepended to the sequence of feature patches. Prompt $P$ is inserted between class token and patches. $W_{emb}^0 \in \mathbb{R}^{(d_P^2 \times d_C) \times d_D}$ is a linear transformation projecting $X_{pat}^0$ into an embedding space, where $d_D$ is the embedding dimension. $X_{pos}$ represents the position embedding. The output of the embedding layer serves as the input for the initial transformer layer. The output of each transformer block is subsequently forwarded to the next transformer block through

$$([CLS^l, Z^l]) = \text{Transformer}_l([CLS^{l-1}, P^{l-1}, Z^{l-1}]). \quad (23)$$

Each transformer layer comprises a multi-head self-attention (MHSA) module and a feed-forward neural network (FFN) module. The output of each transformer block comprises a tensor $P^{(l)}$ at the corresponding position of $P^{l-1}$. However, we adopt the prompt $P^l$ generated by an encoder for the next layer instead of retaining $P^{(l)}$. Namely, $[CLS^l, P^l, Z^l]$ is served as the input of next transformer block.

## 4 Experiments

### 4.1 Experimental Setup

**Benchmarks:** We conduct experiments on various continual learning datasets, including **Split CIFAR-100** [21], **Split DomainNet** [34] and **Split Pets** [33], **Split CropDiseases** [31] and **Split EuroSAT** [14]. Split CIFAR-100 contains 60,000 RGB images over 100

**Table 2: Experimental results (%) on benchmarks from various fields including Split Pets, Split EuroSAT and Split CropDiseases.**

| Method | Split Pets | | | Split EuroSAT | | | Split CropDiseases | | |
|---|---|---|---|---|---|---|---|---|---|
| | Avg Acc ↑ | Lrn Acc ↑ | Forgetting ↓ | Avg Acc ↑ | Lrn Acc ↑ | Forgetting ↓ | Avg Acc ↑ | Lrn Acc ↑ | Forgetting ↓ |
| EWC [20] | 59.40 ± 0.14 | 66.20 ± 0.42 | 8.85 ± 0.35 | 47.40 ± 1.13 | 46.30 ± 3.54 | 2.30 ± 1.56 | 73.30 ± 5.09 | 80.90 ± 7.92 | 9.30 ± 3.25 |
| LwF [24] | 62.50 ± 1.63 | 76.00 ± 1.27 | 18.15 ± 0.92 | 40.40 ± 5.37 | 45.20 ± 3.82 | 6.00 ± 1.84 | 75.10 ± 1.98 | 89.55 ± 2.05 | 27.35 ± 1.95 |
| L2P [50] | 78.34 ± 0.92 | 89.84 ± 0.37 | 15.01 ± 1.10 | 69.17 ± 8.62 | 79.15 ± 3.78 | 12.47 ± 6.05 | 59.73 ± 4.11 | 75.33 ± 5.20 | 12.78 ± 2.83 |
| DualPrompt [49] | 86.85 ± 0.76 | 92.14 ± 0.37 | 8.38 ± 0.74 | 79.41 ± 1.94 | 87.12 ± 0.04 | 12.78 ± 1.23 | 84.23 ± 2.22 | 90.18 ± 1.58 | 7.04 ± 0.81 |
| DAP [17] | 91.02 ± 0.44 | 92.91 ± 0.19 | 1.21 ± 0.45 | 98.18 ± 0.56 | 98.54 ± 0.47 | **0.61 ± 0.53** | 97.88 ± 0.89 | 99.34 ± 0.09 | 1.71 ± 1.02 |
| GPLS | **94.53 ± 0.81** | **95.84 ± 0.92** | **1.61 ± 0.58** | **98.59 ± 0.41** | **99.06 ± 0.55** | 0.64 ± 0.45 | **98.85 ± 0.78** | **99.59 ± 0.61** | **0.87 ± 0.32** |

classes, which is randomly splitted into 10 incremental tasks of disjoint classes. Split DomainNet is a dataset composed of images from 6 distinct types with a total of 345 categories, which is splitted into 15 tasks with each tack containing 23 disjoint classes. Split Pets [33] has 35 categories of pet images with about 200 images per category, which is splitted into 7 tasks. Split CropDiseases [31] contains 35 categories of diseased plant images, which is splitted into 7 tasks. EuroSAT [14] is a collection of satellite images of the landscapes. Split EuroSAT is built by splitting the original 10 classes into 5 tasks of 2 disjoint classes.

**Baselines:** We compare our GPLS with seven prompt-based continual learning methods, including L2P [50], DualPrompt [49], CODA-Prompt [41], S-Prompt [47], LGCL [18], HiDe-Prompt [45] and DAP [17]. In addition, three representative replay-based approaches including BiC [52], DER++ [4] and DyTox [9], two representative regularization-based approaches including EWC [20] and LwF [24] and one dynamic expansion approach ESN [48] adopting ViT as backbone also be considered. All these baselines are continual learning methods designed for class-incremental scenario that task IDs are not available during the testing process.

**Implementation:** We follow the similar implementations as previous work [17]. Specifically, we adopt ViT-B/16 [8] pre-trained on ImageNet as the backbone and Adam ($\beta_1$=0.9, $\beta_2 = 0.9$) with learning rate of 0.01 as the optimizer. All the input images are resized to resolution of $224 \times 224$, normalized range from 0 to 1 and packaged into 128 samples per batch. We utilize an encoder to generate prompts with a fixed length of 10 at each time. To save computational costs, we adopt the simplest one layer MLP encoder [19].

**Evaluation Metrics:** We repeat each experiment over 3 times with different random seeds and report their average values with standard errors using three familiar metrics in CL. (1) **Average accuracy** (Avg Acc ↑) of all the tasks after the model have been trained on the last task $T$, which can be expressed as Avg Acc $= \frac{1}{T} \sum_{t=1}^{T} A_{t,T}$. Here $A_{t,T}$ denotes the test accuracy on task $t$ after learning task $T$. (2) **Learning accuracy** (Lrn Acc): The average accuracy of each task right after the model is trained on the incoming tasks, namely Lrn Acc $= \frac{1}{T} \sum_{t=1}^{T} A_{t,t}$. (3) **Forgetting**: The difference between the maximum knowledge from the previous tasks and the knowledge in the current task, namely $\frac{1}{T-1} \sum_{\tau=1}^{T-1} \max_{t \in \{1,2,...,T-1\}} (A_{\tau,t} - A_{\tau,T})$.

## 4.2 Main Results

The experimental results of GPLS and other continual learning baselines on Split CIFAR-100 and Split DomainNet are shown in Table 1. Note that we take Avg Acc as the primary metric and Forgetting as the secondary indicator. As the first prompt generation method in continual learning, DAP achieved unprecedented performance on Split CIFAR-100 and Split DomainNet by designing the architecture of a prompt generator. Nevertheless, our GPLS adopt an encoder to encode prompts in the latent space and guide the training of encoder through variational Bayesian theory, further improving the performance of prompt generation methods. Specifically, GPLS exhibits the Avg Acc advantages of 2.17% and 6.62% on Split CIFAR-100 and Split DomainNet respectively compared to DAP, which also decreases the forgetting rate by 1.16% and 1.74% on these two datasets.

To demonstrate the excellent domain adaptablity of GPLS, we also conduct experiments on Split Pets, Split EuroSAT, and Split CropDiseases, which belong to the fields of animals, aviation, and agriculture respectively. As shown in Table 2, although DAP has achieved high accuracy on these three datasets, we can still further improve performance to state-of-the-art level. In terms of the most important indicator Avg Acc, GPLS are 3.51%, 0.40%, and 0.96% ahead of DAP on Split Pets, Split EuroSAT and Split CropDiseases respectively. Notably, improving performance on the latter two datasets is challenging due to the already elevated accuracy achieved by DAP. In summary, GPLS achieves consistently better accuracy than baselines, which can be attributed to the latent space theory, encoder design, and variational objective function.

## 4.3 Ablation Study

**Ablation on variables for generating prompts.** From a holistic perspective, the prompts in GPLS are produced by an encoder. However, a more detailed perspective on component level reveals that GPLS is determined by the collaborative influence of three variables including $\mu$, $\Sigma$ and $\epsilon$. Table 3 shows the experimental results after removing each variable separately. (1) Firstly, removing $\mu$ and generating $P$ through $\Sigma \odot \epsilon$ leads to a slight decrease in the overall performance, indicating that the population moment plays an effective role in prompt generation, but its impact is relatively small. (2) Secondly, we remove $\Sigma$ and construct $P$ through $\mu + \epsilon$. The Avg Acc and Lrn Acc of GPLS exhibit significant declines, and the forgetting rate escalates rapidly. These phenomenon indicates that the overall training framework has suffered from severe catastrophic forgetting. Therefore, $\Sigma$ plays a more important role in domain adaptation for prompt generation. In addition, $\mathcal{L}_{nlog\_cov}$ and $\mathcal{L}_{nexp\_Mah}$ are both related to covariance matrix $\Sigma$, which indicates that $\Sigma$ will directly affect the total loss. (3) Thirdly, we ablate $\epsilon$ and utilize $\mu + \Sigma$ to generate prompts. The results show

Chengyi Yang, Wentao Liu, Shisong Chen, Jiayin Qi, & Aimin Zhou

**Table 3: Ablation results (%) of variables for prompt generation on Split CIFAR-100.**

| Method | Avg Acc (↑) | Lrn Acc (↑) | Forgetting (↓) |
|---|---|---|---|
| GPLS | **96.22 ± 0.43** | **97.12 ± 0.51** | **1.12 ± 0.32** |
| Ablate $\mu$ | 95.29 ± 0.63 | 96.73 ± 0.42 | 1.62 ± 0.28 |
| Ablate $\Sigma$ | 68.02 ± 4.15 | 81.93 ± 3.37 | 15.58 ± 2.03 |
| Ablate $\epsilon$ | 90.49 ± 1.21 | 95.55 ± 0.98 | 5.63 ± 0.74 |

**Table 4: Ablation results (%) of the constraints in ELBO on Split CIFAR-100.**

| Method | Avg Acc (↑) | Lrn Acc (↑) | Forgetting (↓) |
|---|---|---|---|
| GPLS | **96.22 ± 0.43** | **97.12 ± 0.51** | **1.12 ± 0.32** |
| w/o $\mathcal{L}_{nexp\_Mah}$ | 95.07 ± 1.23 | 96.51 ± 0.81 | 1.89 ± 0.72 |
| w/o $\mathcal{L}_{nlog\_cov}$ | 95.56 ± 1.08 | 96.83 ± 0.62 | 1.67 ± 0.63 |
| w/o $\mathcal{L}_{nexp\_Mah}$ & $\mathcal{L}_{nlog\_cov}$ | 94.98 ± 1.26 | 96.32 ± 2.25 | 2.11 ± 1.02 |

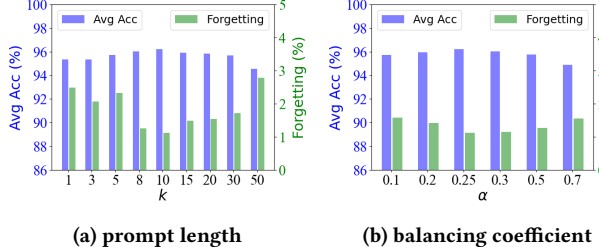

**(a) prompt length**      **(b) balancing coefficient**

**Figure 4: Comparison of Avg Acc (%) and Forgetting (%) with different settings of prompt length and balancing coefficient on Split CIFAR-100.**

that GPLS also experience a decrease in accuracy and an increase in forgetting, which also verifies the effectiveness of $\epsilon$.

**Ablation on ELBO constraint.** We also conducted ablation experiments on $\mathcal{L}_{nexp\_Mah}$ and $\mathcal{L}_{nlog\_cov}$. The results in Table 4 show that removing any part of the constraints will result in inferior accuracy and higher forgetting. Specifically, deleting $\mathcal{L}_{nexp\_Mah}$ and $\mathcal{L}_{nlog\_cov}$ would result in a decrease of 1.15% and 0.66% in Avg Acc respectively, which also show that $\mathcal{L}_{nexp\_Mah}$ plays a more important role in the sense of improving performance compared to $\mathcal{L}_{nlog\_cov}$.

### 4.4 Hyperparameter Analysis

**Analysis on prompt length $k$.** To investigate the impact of variation on prompt length $k$ for model performance, we set the value of $k$ as {1, 3, 5, 8, 10, 15, 20, 30, 50} successively. Table 4(a) reports Avg Acc and Forgetting when GPLS are performed on Split CIFAR-100. We observed that when the prompt length equals 10, the accuracy and forgetting reach the optimal level simultaneously.

**Analysis on balancing coefficient $\alpha$.** The hyperparameter $\alpha$ is applied to balance the importance between matching loss and our variational loss. We change its value as {0.1, 0.2, 0.25, 0.3, 0.5, 0.7} successively. As illustrated in Table 4(b), our method achieves the best performance when $\alpha = 0.25$.

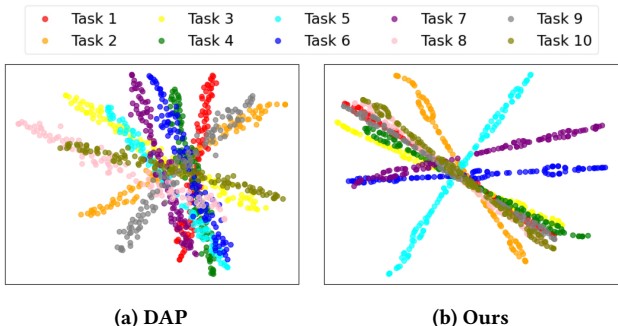

**(a) DAP**        **(b) Ours**

**Figure 5: T-SNE visualizations of generated prompts for 10 tasks on Split CIFAR-100. Note that the prompt distribution in GPLS refers to all possible prompts following the prompt distribution, and one generated prompt can be viewed as an observation. In addition, prompts are not utilized to predict classification labels like logits, therefore overlapping parts under different tasks are allowed due to the varying levels of similarity between these tasks.**

### 4.5 T-SNE Visualization for Prompt Comparison

Since both DAP and GPLS are prompt generation methods rather than prompt optimization methods, we compare the prompts generated by our GPLS with those generated by DAP in Figure 5. Although DAP demonstrates effective domain adaptability through prompt generation, its T-SNE plot reveals challenges in identifying distinct boundaries between the prompts it generates and lacks a unique data center. In contrast, prompts generated by GPLS exhibit a discernible center across various tasks, which indicates that GPLS can effectively identify the commonalities between different tasks. Moreover, prompts in GPLS display clearer boundaries between different tasks, which indicates that GPLS can clearly understand the differences between different tasks and has better domain adaptability.

## 5 Conclusion

In this paper, we analyze that the trainable parameters in prompt generation methods have the property of localization, centralization, and indirectness, which poses challenges in achieving higher performance. In addition, we note that the prompts generated by a prompt generator are uncertain and exert an implicit influence on classification results, resembling the concept of latent variables in Bayesian learning. Motivated by this insight, we propose a method called generating prompts in latent space for rehearsal-free continual learning based on variational Bayesian theory. We design a variational encoder which encodes task information and feature representation into prompts in latent space with reparameterization technique. We insert the prompts into the inputs of transformer blocks, and train the encoder by minimizing the negative evidence lower bound. We conduct extensive experiments to verify the effectiveness of our GPLS, and the results demonstrate that GPLS consistently achieves the state-of-the-art performance on five continual learning benchmarks.

## Acknowledgments

This work is supported by National Natural Science Foundation of China (No. 72293583, No. 72293580), Science and Technology Commission of Shanghai Municipality Grant (No. 22511105901) and Sino-German Research Network (GZ570).

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
