# OpenReview forum: "Generating Prompts in Latent Space for Rehearsal-free Continual Learning"
_acmmm.org/ACMMM/2024/Conference — MM2024 Poster_

### Official Review · Reviewer_27NA · 2024-05-24

**Rating:** 4
**Confidence:** 2

**Summary:**

This paper proposes GPLS which generates prompts by an encoder with specific tsak-level tokens and feature information. The paper is well organized with good quality.

**Strengths:**

The proposed method is well designed with good results.

**Limitations:**

As we know, DAP is very sensitive to the batch-wise or  instance-wise match setups in the evaluation stage.  The key difference between the two setups lies in the predicting the task ID for instances in a batch. In the former, a single task ID is predicted for the entire batch, while in the latter, the task IDs are predicted on a per-instance basis.

The batch-wise setup has a correcting step which is not realistic in real applications. The good performance of the proposed method is still evaluated in the batch-wise setup.  However, the proposed method should also be evaluated in the instance-wise setup which is more challenging without a high matching accuracy.

**Suitability:**

2

---

### Official Review · Reviewer_gfGG · 2024-05-24

**Rating:** 3
**Confidence:** 4

**Summary:**

This paper tries to provide guidance in training the prompt generator in prompt-based continual learning methods, and propose Generating Prompts in Latent Space (GPLS). utilizes an encoder to encode task and feature information into prompts and provides enhanced, targeted guidance for the training process through variational inference. The experimental results show that the GPLS prompts show clearer boundaries between different tasks, enable a clear understanding of the differences between tasks, and have better domain adaptation.

**Strengths:**

This paper is well organized and easy to follow. The experimental results are good.

**Limitations:**

1. The authors claim their main focus is “…, existing methods… neglecting the importance of providing effective guidance for training the generator”. However, they have not illustrated this “neglected guidance” problem, and there is no experiment to reflect this point.
2. How do you convert Problem (4) into Problem (9)? Typically, the ELBO is one component of a problem. This problem can usually be split into two distinct parts: one is ELBO and the other is the KL divergence. Taking Equation (6) as an example, we can divide Problem (6) into two separate parts: the ELBO(q) and the KL divergence. Maximizing ELBO(q) equals minimizing the log probability, logp(Y|X). Looking at it this way, the authors suggest that Problem (4) is equivalent to Problem (6). However, I believe this equivalency is either incorrect or somewhat unusual. Please provide a more detailed analysis of this interpretation.
3. How do you get or define the Eq.(9) of $p(P,Y|X)$. In our common sense, $p(P, Y|X)$ informally means: inferring $P$ and $Y$, given $X$. However, the authors consider $P$ and $X$ as inputs, which means inferring $Y$, given $P$ and $X$. It seems unusual or possibly incorrect, please provide a thorough analysis of this. A similar issue also exists in Eq.(11).
4. How to solve the forgetting of the prompt-generator’s forgetting? The main concern of continual learning is to alleviate the forgetting problem of the trainable network, but there is no strategy for this point in the method section after carefully reviewing it.
5. Typos: (1) “Where …” after the equation should not be capitalized, it is a clause of one full sentence. (2) there is “-” left of the KL in the Eq.(7) of Supplementary Material; (3) $d_D$ in Line 612 is undefined and wrong used; (4) suggest replacing “Figure 4a” with “Figure 4(a)” in Line 811.

Please clarify the mentioned concerns, as it is important for the reviewer to update the final decision.

**Suitability:**

3

---

### Official Review · Reviewer_dQS5 · 2024-05-24

**Rating:** 4
**Confidence:** 3

**Summary:**

This paper introduces Generating Prompts in Latent Space (GPLS), which considers prompts as latent variables to account for the uncertainty of prompt generation and aligns with the fact that prompts are inserted into the hidden layer outputs and exert an implicit influence on classification. GPLS adopts a trainable encoder to encode task and feature information into prompts with reparameterization technique, and provides refined and targeted guidance for the training process through the evidence lower bound (ELBO) related to Mahalanobis distance.

**Strengths:**

1. The paper is well organized
2. The formulas and symbols in the article are explained in detail

**Limitations:**

1. On line 447 of the paper, the authors assume that q(P) is a multivariate Gaussian distribution. How reasonable is this assumption? If q(P) does not follow a Gaussian distribution, what would the loss function look like?
2. How can it be ensured that GPLS can still generate correct prompts for old task samples as new tasks are updated?
3. Sections 3.3 and 3.4 are almost common derivations in variational inference. Could the authors explain in detail what the main contribution of this part is?

**Suitability:**

2

---

### Official Review · Reviewer_5Yev · 2024-05-25

**Rating:** 5
**Confidence:** 4

**Summary:**

The paper introduces a novel method called Generating Prompts in Latent Space (GPLS) for rehearsal-free continual learning. The core idea is to enhance prompt-based continual learning by treating prompts as latent variables, thereby capturing the uncertainty in prompt generation. GPLS uses a trainable encoder to encode task and feature information into prompts using a reparameterization technique. This encoder generates prompts in the latent space, making it more adaptable to different tasks. The training process is guided by a variational loss function derived from the Evidence Lower Bound (ELBO), specifically related to Mahalanobis distance. Extensive experiments demonstrate that GPLS achieves state-of-the-art performance on various continual learning benchmarks, including Split CIFAR-100 and Split DomainNet.

**Strengths:**

- The paper provides a solid theoretical foundation by leveraging variational inference and the Evidence Lower Bound (ELBO). This approach is mathematically sound and aligns with the principles of probabilistic modeling, enhancing the credibility and robustness of the proposed method.
- The method is technically sound, with a well-defined problem formulation and solution. The use of a trainable encoder and the reparameterization technique ensures that the generated prompts are effective and adaptable.
- The paper is well-organized and clearly written.

**Limitations:**

- The paper adopts a simple one-layer MLP encoder to generate prompts, which may not fully capture the complex interactions and dependencies in high-dimensional data. In other words, the prompt generation process might be overly simplistic, potentially making the proposed method not fundamentally different from directly optimizing the prompts.
- The paper does not thoroughly address the computational overhead and model size comparison. Detailed analysis and comparisons of computational requirements and scalability with other state-of-the-art methods are lacking.
- The method still relies on predicting task IDs to generate prompts, which is time-consuming and makes the approach similar to prompt selection strategies. This reliance reduces the distinction between prompt generation and prompt selection methods, potentially diminishing the novelty of the proposed approach.

**Suitability:**

3

---

### Meta-Review · Area_Chair_DFa7 · 2024-07-03

**Recommendation:** Accept (Poster)
**Confidence:** 5

**Metareview:**

The strengths of this paper include a generally well-structured presentation and novel ideas. However, some weaknesses need to be addressed: the presentation of technical details needs improvement, as pointed out by all the reviewers. This includes aspects such as probabilistic modeling, the derivation/formulation of ELBO, and the model design related to the task ID. The authors should address these minor but essential issues in the next version.